# ON DISTRIBUTED ADAPTIVE OPTIMIZATION WITH GRADIENT COMPRESSION

**Xiaoyun Li, Belhal Karimi, Ping Li**

Cognitive Computing Lab
Baidu Research
10900 NE 8th St. Bellevue, WA 98004, USA
{xiaoyunli,belhalkarimi,liping11}@baidu.com

## ABSTRACT

We study COMP-AMS, a distributed optimization framework based on gradient averaging and adaptive AMSGrad algorithm. Gradient compression with error feedback is applied to reduce the communication cost in the gradient transmission process. Our convergence analysis of COMP-AMS shows that such compressed gradient averaging strategy yields same convergence rate as standard AMSGrad, and also exhibits the linear speedup effect w.r.t. the number of local workers. Compared with recently proposed protocols on distributed adaptive methods, COMP-AMS is simple and convenient. Numerical experiments are conducted to justify the theoretical findings, and demonstrate that the proposed method can achieve same test accuracy as the full-gradient AMSGrad with substantial communication savings. With its simplicity and efficiency, COMP-AMS can serve as a useful distributed training framework for adaptive gradient methods.

## 1 INTRODUCTION

Deep neural network has achieved the state-of-the-art learning performance on numerous AI applications, e.g., computer vision and natural language processing (Graves et al., 2013; Goodfellow et al., 2014; He et al., 2016; Young et al., 2018; Zhang et al., 2018), reinforcement learning (Mnih et al., 2013; Levine et al., 2016; Silver et al., 2017), recommendation systems (Covington et al., 2016), computational advertising (Zhao et al., 2019; Xu et al., 2021; Zhao et al., 2022), etc. With the increasing size of data and growing complexity of deep neural networks, standard single-machine training procedures encounter at least two major challenges:

- Due to the limited computing power of a single-machine, processing the massive number of data samples takes a long time—training is too slow. Many real-world applications cannot afford spending days or even weeks on training.

- In many scenarios, data are stored on multiple servers, possibly at different locations, due to the storage constraints (massive user behavior data, Internet images, etc.) or privacy reasons (Chang et al., 2018). Hence, transmitting data among servers might be costly.

*Distributed learning* framework has been commonly used to tackle the above two issues. Consider the distributed optimization task where $n$ workers jointly solve the following optimization problem

$$\min_\theta f(\theta) := \min_\theta \frac{1}{n} \sum_{i=1}^n f_i(\theta) = \frac{1}{n} \sum_{i=1}^n \mathbb{E}_{x \sim \mathcal{X}_i}[F_i(\theta; x)], \tag{1}$$

where the non-convex function $f_i$ represents the average loss over the local data samples for worker $i \in [n]$, and $\theta \in \mathbb{R}^d$ the global model parameter. $\mathcal{X}_i$ is the data distribution on each local node. In the classical centralized distributed setting, in each iteration the central server uniformly randomly assigns the data to $n$ local workers ($\mathcal{X}_i$'s are the same), at which the gradients of the model are computed in parallel. Then the central server aggregates the local gradients, updates the global model (e.g., by stochastic gradient descent (SGD)), and transmits back the updated model to the

local nodes for subsequent gradient computation. The scenario where $\mathcal{X}_i$'s are different gives rise to the recently proposed Federated Learning (FL) (McMahan et al., 2017) framework, which will not be the major focus of this work. As we can see, distributed training naturally solves aforementioned issues: 1) We use $n$ computing nodes to train the model, so the time per training epoch can be largely reduced; 2) There is no need to transmit the local data to central server. Besides, distributed training also provides stronger error tolerance since the training process could continue even one local machine breaks down. As a result of these advantages, there has been a surge of study and applications on distributed systems (Nedic & Ozdaglar, 2009; Boyd et al., 2011; Duchi et al., 2012; Goyal et al., 2017; Hong et al., 2017; Koloskova et al., 2019; Lu et al., 2019).

**Gradient compression.** Among many optimization strategies, SGD is still the most popular prototype in distributed training for its simplicity and effectiveness (Chilimbi et al., 2014; Agarwal et al., 2018; Mikami et al., 2018). Yet, when the deep learning model is very large, the communication between local nodes and central server could be expensive, and the burdensome gradient transmission would slow down the whole training system. Thus, reducing the communication cost in distributed SGD has become an active topic, and an important ingredient of large-scale distributed systems (e.g., Seide et al. (2014)). Solutions based on quantization, sparsification and other compression techniques of the local gradients have been proposed, e.g., Aji & Heafield (2017); Alistarh et al. (2017); Sa et al. (2017); Wen et al. (2017); Bernstein et al. (2018); Stich et al. (2018); Wangni et al. (2018); Ivkin et al. (2019); Yang et al. (2019); Haddadpour et al. (2020). However, it has been observed both theoretically and empirically (Stich et al., 2018; Ajalloeian & Stich, 2020), that directly updating with the compressed gradients usually brings non-negligible performance downgrade in terms of convergence speed and accuracy. To tackle this problem, studies (e.g., Stich et al. (2018); Karimireddy et al. (2019)) show that the technique of *error feedback* can to a large extent remedy the issue of such gradient compression, achieving the same convergence rate as full-gradient SGD.

**Adaptive optimization.** In recent years, adaptive optimization algorithms (e.g., AdaGrad (Duchi et al., 2010), Adam (Kingma & Ba, 2015) and AMSGrad (Reddi et al., 2018)) have become popular because of their superior empirical performance. These methods use different implicit learning rates for different coordinates that keep changing adaptively throughout the training process, based on the learning trajectory. In many cases, adaptive methods have been shown to converge faster than SGD, sometimes with better generalization as well. Nevertheless, the body of literature that extends adaptive methods to distributed training is still fairly limited. In particular, even the simple gradient averaging approach, though appearing standard, has not been analyzed for adaptive optimization algorithms. Given that distributed SGD with compressed gradient averaging can match the performance of standard SGD, one natural question is: is it also true for adaptive methods? In this work, we fill this gap formally, by analyzing COMP-AMS, a distributed adaptive optimization framework using the gradient averaging protocol, with communication-efficient gradient compression. Our method has been implemented in the PaddlePaddle platform (`www.paddlepaddle.org.cn`).

**Our contributions.** We study a simple algorithm design leveraging the *adaptivity* of AMSGrad and the computational virtue of *local gradient compression*:

- We propose COMP-AMS, a synchronous distributed adaptive optimization framework based on global averaging with gradient compression, which is efficient in both communication and memory as no local moment estimation is needed. We consider the BlockSign and Top-$k$ compressors, coupled with the error-feedback technique to compensate for the bias implied by the compression step for fast convergence.

- We provide the convergence analysis of distributed COMP-AMS (with $n$ workers) in smooth non-convex optimization. In the special case of $n = 1$ (single machine), similar to SGD, gradient compression with error feedback in adaptive method achieves the same convergence rate $\mathcal{O}(\frac{1}{\sqrt{T}})$ as the standard full-gradient counterpart. Also, we show that with a properly chosen learning rate, COMP-AMS achieves $\mathcal{O}(\frac{1}{\sqrt{nT}})$ convergence, implying a linear speedup in terms of the number of local workers to attain a stationary point.

- Experiments are conducted on various training tasks on image classification and sentiment analysis to validate our theoretical findings on the linear speedup effect. Our results show that COMP-AMS has comparable performance with other distributed adaptive methods, and approaches the accuracy of full-precision AMSGrad with a substantially reduced communication cost. Thus, it can serve as a convenient distributed training strategy in practice.

## 2 RELATED WORK

### 2.1 DISTRIBUTED SGD WITH COMPRESSED GRADIENTS

**Quantization.** To reduce the expensive communication in large-scale distributed SGD training systems, extensive works have considered various compression techniques applied to the gradient transaction procedure. The first strategy is quantization. Dettmers (2016) condenses 32-bit floating numbers into 8-bits when representing the gradients. Seide et al. (2014); Bernstein et al. (2018; 2019); Karimireddy et al. (2019) use the extreme 1-bit information (sign) of the gradients, combined with tricks like momentum, majority vote and memory. Other quantization-based methods include QSGD (Alistarh et al., 2017; Zhang et al., 2017; Wu et al., 2018) and LPC-SVRG (Yu et al., 2019b), leveraging unbiased stochastic quantization. Quantization has been successfully applied to industrial-level applications, e.g., Xu et al. (2021). The saving in communication of quantization methods is moderate: for example, 8-bit quantization reduces the cost to 25% (compared with 32-bit full-precision). Even in the extreme 1-bit case, the largest compression ratio is around $1/32 \approx 3.1\%$.

**Sparsification.** Gradient sparsification is another popular solution which may provide higher compression rate. Instead of commuting the full gradient, each local worker only passes a few coordinates to the central server and zeros out the others. Thus, we can more freely choose higher compression ratio (e.g., 1%, 0.1%), still achieving impressive performance in many applications (Lin et al., 2018). Stochastic sparsification methods, including uniform and magnitude based sampling (Wangni et al., 2018), select coordinates based on some sampling probability, yielding unbiased gradient compressors with proper scaling. Deterministic methods are simpler, e.g., Random-$k$, Top-$k$ (Stich et al., 2018; Shi et al., 2019) (selecting $k$ elements with largest magnitude), Deep Gradient Compression (Lin et al., 2018), but usually lead to biased gradient estimation. More applications and analysis of compressed distributed SGD can be found in Alistarh et al. (2018); Jiang & Agrawal (2018); Jiang et al. (2018); Shen et al. (2018); Basu et al. (2019), among others.

**Error Feedback (EF).** Biased gradient estimation, which is a consequence of many aforementioned methods (e.g., signSGD, Top-$k$), undermines the model training, both theoretically and empirically, with slower convergence and worse generalization (Ajalloeian & Stich, 2020; Beznosikov et al., 2020). The technique of *error feedback* is able to "correct for the bias" and fix the convergence issues. In this procedure, the difference between the true stochastic gradient and the compressed one is accumulated locally, which is then added back to the local gradients in later iterations. Stich et al. (2018); Karimireddy et al. (2019) prove the $\mathcal{O}(\frac{1}{T})$ and $\mathcal{O}(\frac{1}{\sqrt{T}})$ convergence rate of EF-SGD in strongly convex and non-convex setting respectively, matching the rates of vanilla SGD (Nemirovski et al., 2009; Ghadimi & Lan, 2013). More recent works on the convergence rate of SGD with error feedback include Stich & Karimireddy (2019); Zheng et al. (2019); Richtárik et al. (2021), etc.

### 2.2 ADAPTIVE OPTIMIZATION

In each SGD update, all the coordinates share the same learning rate, which is either constant or decreasing through the iterations. Adaptive optimization methods cast different learning rates on each dimension. For instance, AdaGrad, developed in Duchi et al. (2010), divides the gradient elementwise by $\sqrt{\sum_{t=1}^{T} g_t^2} \in \mathbb{R}^d$, where $g_t \in \mathbb{R}^d$ is the gradient vector at time $t$ and $d$ is the model dimensionality. Thus, it intrinsically assigns different learning rates to different coordinates throughout the training— elements with smaller previous gradient magnitudes tend to move a larger step via larger learn-

---

**Algorithm 1** AMSGRAD (Reddi et al., 2018)

1: **Input**: parameters $\beta_1$, $\beta_2$, $\epsilon$, learning rate $\eta_t$
2: **Initialize:** $\theta_1 \in \mathbb{R}^d$, $m_0 = v_0 = \mathbf{0} \in \mathbb{R}^d$
3: **for** $t = 1, \ldots, T$ **do**
4:     Compute stochastic gradient $g_t$ at $\theta_t$
5:     $m_t = \beta_1 m_{t-1} + (1 - \beta_1) g_t$
6:     $v_t = \beta_2 v_{t-1} + (1 - \beta_2) g_t^2$
7:     $\hat{v}_t = \max(\hat{v}_{t-1}, v_t)$
8:     $\theta_{t+1} = \theta_t - \eta_t \frac{m_t}{\sqrt{\hat{v}_t} + \epsilon}$
9: **end for**

---

ing rate. Other adaptive methods include AdaDelta (Zeiler, 2012) and Adam (Kingma & Ba, 2015), which introduce momentum and moving average of second moment estimation into AdaGrad hence leading to better performances. AMSGrad (Reddi et al., 2018) (Algorithm 1, which is the prototype in our paper), fixes the potential convergence issue of Adam. Wang et al. (2021) and Zhou et al.

(2020) improve the convergence and generalization of AMSGrad through optimistic acceleration and differential privacy.

Adaptive optimization methods have been widely used in training deep learning models in language, computer vision and advertising applications, e.g., Choi et al. (2019); You et al. (2020); Zhang et al. (2021); Zhao et al. (2022). In distributed setting, Nazari et al. (2019); Chen et al. (2021b) study decentralized adaptive methods, but communication efficiency was not considered. Mostly relevant to our work, Chen et al. (2021a) proposes a distributed training algorithm based on Adam, which requires every local node to store a local estimation of the moments of the gradient. Thus, one has to keep extra two more tensors of the model size on each local worker, which may be less feasible in terms of memory particularly with large models. More recently, Tang et al. (2021) proposes an Adam pre-conditioned momentum SGD method. Chen et al. (2020); Karimireddy et al. (2020); Reddi et al. (2021) proposed local/global adaptive FL methods, which can be further accelerated via layer-wise adaptivity (Karimi et al., 2021).

## 3 COMMUNICATION-EFFICIENT ADAPTIVE OPTIMIZATION

### 3.1 GRADIENT COMPRESSORS

In this paper, we mainly consider deterministic $q$-deviate compressors defined as below.

**Assumption 1.** *The gradient compressor $\mathcal{C} : \mathbb{R}^d \mapsto \mathbb{R}^d$ is $q$-deviate: for $\forall x \in \mathbb{R}^d$, $\exists\, 0 \leq q < 1$ such that $\|\mathcal{C}(x) - x\| \leq q \|x\|$.*

Larger $q$ indicates heavier compression, while smaller $q$ implies better approximation of the true gradient. $q = 0$ implies $\mathcal{C}(x) = x$, i.e., no compression. In the following, we give two popular and efficient $q$-deviate compressors that will be adopted in this paper.

**Definition 1** (Top-$k$). *For $x \in \mathbb{R}^d$, denote $\mathcal{S}$ as the size-$k$ set of $i \in [d]$ with largest $k$ magnitude $|x_i|$. The **Top-$k$** compressor is defined as $\mathcal{C}(x)_i = x_i$, if $i \in \mathcal{S}$; $\mathcal{C}(x)_i = 0$ otherwise.*

**Definition 2** (Block-Sign). *For $x \in \mathbb{R}^d$, define $M$ blocks indexed by $\mathcal{B}_i$, $i = 1, ..., M$, with $d_i := |\mathcal{B}_i|$. The **Block-Sign** compressor is defined as $\mathcal{C}(x) = [sign(x_{\mathcal{B}_1})\frac{\|x_{\mathcal{B}_1}\|_1}{d_1}, ..., sign(x_{\mathcal{B}_M})\frac{\|x_{\mathcal{B}_M}\|_1}{d_M}]$, where $x_{\mathcal{B}_i}$ is the sub-vector of $x$ at indices $\mathcal{B}_i$.*

**Remark 1.** *It is well-known (Stich et al., 2018) that for **Top-$k$**, $q^2 = 1 - \frac{k}{d}$. For **Block-Sign**, by Cauchy-Schwartz inequality we have $q^2 = 1 - \min_{i \in [M]} \frac{1}{d_i}$ where $M$ and $d_i$ are defined in Definition 2 (Zheng et al., 2019).*

The intuition behind **Top-$k$** is that, it has been observed empirically that when training many deep models, most gradients are typically very small, and gradients with large magnitude contain most information. The **Block-Sign** compressor is a simple extension of the 1-bit **SIGN** compressor (Seide et al., 2014; Bernstein et al., 2018), adapted to different gradient magnitude in different blocks, which, for neural nets, are usually set as the distinct network layers. The scaling factor in Definition 2 is to preserve the (possibly very different) gradient magnitude in each layer. In principle, **Top-$k$** would perform the best when the gradient is effectively sparse, while **Block-Sign** compressor is favorable by nature when most gradients have similar magnitude within each layer.

### 3.2 COMP-AMS: DISTRIBUTED ADAPTIVE TRAINING BY GRADIENT AGGREGATION

We present in Algorithm 2 the proposed communication-efficient distributed adaptive method in this paper, COMP-AMS. This framework can be regarded as an analogue to the standard synchronous distributed SGD: in each iteration, each local worker transmits to the central server the compressed stochastic gradient computed using local data. Then the central server takes the average of local gradients, and performs an AMSGrad update. In Algorithm 2, lines 7-8 depict the error feedback operation at local nodes. $e_{t,i}$ is the accumulated error from gradient compression on the $i$-th worker up to time $t - 1$. This residual is added back to $g_{t,i}$ to get the "corrected" gradient. In Section 4 and Section 5, we will show that error feedback, similar to the case of SGD, also brings good convergence behavior under gradient compression in distributed AMSGrad.

---

**Algorithm 2** Distributed COMP-AMS with error feedback (EF)

---

1: **Input**: parameters $\beta_1$, $\beta_2$, $\epsilon$, learning rate $\eta_t$
2: **Initialize**: central server parameter $\theta_1 \in \mathbb{R}^d \subseteq \mathbb{R}^d$; $e_{1,i} = \mathbf{0}$ the error accumulator for each worker; $m_0 = \mathbf{0}$, $v_0 = \mathbf{0}$, $\hat{v}_0 = \mathbf{0}$
3: **for** $t = 1, \ldots, T$ **do**
4:     **parallel for worker** $i \in [n]$ **do**:
5:         Receive model parameter $\theta_t$ from central server
6:         Compute stochastic gradient $g_{t,i}$ at $\theta_t$
7:         Compute the compressed gradient $\tilde{g}_{t,i} = \mathcal{C}(g_{t,i} + e_{t,i})$
8:         Update the error $e_{t+1,i} = e_{t,i} + g_{t,i} - \tilde{g}_{t,i}$
9:         Send $\tilde{g}_{t,i}$ back to central server
10:     **end parallel**
11:     **Central server do:**
12:     $\bar{g}_t = \frac{1}{n} \sum_{i=1}^n \tilde{g}_{t,i}$
13:     $m_t = \beta_1 m_{t-1} + (1 - \beta_1)\bar{g}_t$
14:     $v_t = \beta_2 v_{t-1} + (1 - \beta_2)\bar{g}_t^2$
15:     $\hat{v}_t = \max(v_t, \hat{v}_{t-1})$
16:     Update the global model $\theta_{t+1} = \theta_t - \eta_t \frac{m_t}{\sqrt{\hat{v}_t} + \epsilon}$
17: **end for**

---

**Comparison with related methods.** Next, we discuss the differences between COMP-AMS and two recently proposed methods also trying to solve compressed distributed adaptive optimization.

- **Comparison with Chen et al. (2021a).** Chen et al. (2021a) develops a quantized variant of Adam (Kingma & Ba, 2015), called "QAdam". In this method, each worker keeps a local copy of the moment estimates, commonly noted $m$ and $v$, and compresses and transmits the ratio $\frac{m}{v}$ as a whole to the server. Their method is hence very much like the compressed distributed SGD, with the exception that the ratio $\frac{m}{v}$ plays the role of the gradient vector $g$ communication-wise. Thus, two local moment estimators are additionally required, which have same size as the deep learning model. In our COMP-AMS, the moment estimates $m$ and $v$ are kept and updated only at the central server, thus not introducing any extra variable (tensor) on local nodes during training (except for the error accumulator). Hence, COMP-AMS is not only effective in communication reduction, but also efficient in terms of memory (space), which is feasible when training large-scale learners like BERT and CTR prediction models, e.g., Devlin et al. (2019); Xu et al. (2021), to lower the hardware consumption in practice. Additionally, the convergence rate in Chen et al. (2021a) does not improve linearly with $n$, while we prove the linear speedup effect of COMP-AMS.

- **Comparison with Tang et al. (2021)** The recent work (Tang et al., 2021) proposes "1BitAdam". They first run some warm-up training steps using standard Adam, and then store the second moment moving average $v$. Then, distributed Adam training starts with $v$ frozen. Thus, 1BitAdam is actually more like a distributed momentum SGD with some pre-conditioned coordinate-wise learning rates. The number of warm-up steps also needs to be carefully tuned, otherwise bad pre-conditioning may hurt the learning performance. Our COMP-AMS is simpler, as no pre-training is needed. Also, 1BitAdam requires extra tensors for $m$ locally, while COMP-AMS does not need additional local memory.

## 4   CONVERGENCE ANALYSIS

For the convergence analysis of COMP-AMS we will make following additional assumptions.

**Assumption 2.** *(Smoothness) For $\forall i \in [n]$, $f_i$ is $L$-smooth: $\|\nabla f_i(\theta) - \nabla f_i(\vartheta)\| \leq L \|\theta - \vartheta\|$.*

**Assumption 3.** *(Unbiased and bounded stochastic gradient) For $\forall t > 0$, $\forall i \in [n]$, the stochastic gradient is unbiased and uniformly bounded: $\mathbb{E}[g_{t,i}] = \nabla f_i(\theta_t)$ and $\|g_{t,i}\| \leq G$.*

**Assumption 4.** *(Bounded variance) For $\forall t > 0$, $\forall i \in [n]$: (i) the **local variance** of the stochastic gradient is bounded: $\mathbb{E}[\|g_{t,i} - \nabla f_i(\theta_t)\|^2] < \sigma^2$; (ii) the **global variance** is bounded by $\frac{1}{n} \sum_{i=1}^n \|\nabla f_i(\theta_t) - \nabla f(\theta_t)\|^2 \leq \sigma_g^2$.*

In Assumption 3, the uniform bound on the stochastic gradient is common in the convergence analysis of adaptive methods, e.g., Reddi et al. (2018); Zhou et al. (2018); Chen et al. (2019). The global variance bound $\sigma_g^2$ in Assumption 4 characterizes the difference among local objective functions, which, is mainly caused by different local data distribution $\mathcal{X}_i$ in (1). In classical distributed setting where all the workers can access the same dataset and local data are assigned randomly, $\sigma_g^2 \equiv 0$. While typical federated learning (FL) setting with $\sigma_g^2 > 0$ is not the focus of this present paper, we consider the global variance in our analysis to shed some light on the impact of non-i.i.d. data distribution in the federated setting for broader interest and future investigation.

We derive the following general convergence rate of COMP-AMS in the distributed setting. The proof is deferred to Appendix B.

**Theorem 1.** *Denote* $C_0 = \sqrt{\frac{4(1+q^2)^3}{(1-q^2)^2}G^2 + \epsilon}$, $C_1 = \frac{\beta_1}{1-\beta_1} + \frac{2q}{1-q^2}$, $\theta^* = \arg\min f(\theta)$ *defined as* *(1). Under Assumptions 1 to 4, with* $\eta_t = \eta \le \frac{\epsilon}{3C_0\sqrt{2L\max\{2L,C_1\}}}$, *Algorithm 2 satisfies*

$$\frac{1}{T}\sum_{t=1}^{T}\mathbb{E}[\|\nabla f(\theta_t)\|^2] \le 2C_0\Big(\frac{\mathbb{E}[f(\theta_1) - f(\theta^*)]}{T\eta} + \frac{\eta L\sigma^2}{n\epsilon} + \frac{3\eta^2 LC_0 C_1^2 \sigma^2}{n\epsilon^2}$$
$$+ \frac{12\eta^2 q^2 LC_0 \sigma_g^2}{(1-q^2)^2\epsilon^2} + \frac{(1+C_1)G^2 d}{T\sqrt{\epsilon}} + \frac{\eta(1+2C_1)C_1 LG^2 d}{T\epsilon}\Big).$$

The LHS of Theorem 1 is the expected squared norm of the gradient from a uniformly chosen iterate $t \in [T]$, which is a common convergence measure in non-convex optimization. From Theorem 1, we see that the more compression we apply to the gradient vectors (i.e., larger $q$), the larger the gradient magnitude is, i.e., the slower the algorithm converges. This is intuitive as heavier compression loses more gradient information which would slower down the learner to find a good solution.

Note that, COMP-AMS with $n = 1$ naturally reduces to the single-machine (sequential) AMSGrad (Algorithm 1) with compressed gradients instead of full-precision ones. Karimireddy et al. (2019) specifically analyzed this case for SGD, showing that compressed single-machine SGD with error feedback has the same convergence rate as vanilla SGD using full gradients. In alignment with the conclusion in Karimireddy et al. (2019), for adaptive AMSGrad, we have a similar result.

**Corollary 1.** *When* $n = 1$, *under Assumption 1 to Assumption 4, setting the stepsize as* $\eta = \min\{\frac{\epsilon}{3C_0\sqrt{2L\max\{2L,C_1\}}}, \frac{1}{\sqrt{T}}\}$, *Algorithm 2 satisfies*

$$\frac{1}{T}\sum_{t=1}^{T}\mathbb{E}[\|\nabla f(\theta_t)\|^2] \le \mathcal{O}\Big(\frac{1}{\sqrt{T}} + \frac{\sigma^2}{\sqrt{T}} + \frac{d}{T}\Big).$$

Corollary 1 states that with error feedback, single machine AMSGrad with biased compressed gradients can also match the convergence rate $\mathcal{O}(\frac{1}{\sqrt{T}} + \frac{d}{T})$ of standard AMSGrad (Zhou et al., 2018) in non-convex optimization. It also achieves the same rate $\mathcal{O}(\frac{1}{\sqrt{T}})$ of vanilla SGD (Karimireddy et al., 2019) when $T$ is sufficiently large. In other words, error feedback also fixes the convergence issue of using compressed gradients in AMSGrad.

**Linear Speedup.** In Theorem 1, the convergence rate is derived by assuming a constant learning rate. By carefully choosing a decreasing learning rate dependent on the number of workers, we have the following simplified statement.

**Corollary 2.** *Under the same setting as Theorem 1, set* $\eta = \min\{\frac{\epsilon}{3C_0\sqrt{2L\max\{2L,C_1\}}}, \frac{\sqrt{n}}{\sqrt{T}}\}$. *The* COMP-AMS *iterates admit*

$$\frac{1}{T}\sum_{t=1}^{T}\mathbb{E}[\|\nabla f(\theta_t)\|^2] \le \mathcal{O}\Big(\frac{1}{\sqrt{nT}} + \frac{\sigma^2}{\sqrt{nT}} + \frac{n(\sigma^2 + \sigma_g^2)}{T}\Big). \tag{2}$$

In Corollary 2, we see that the global variance $\sigma_g^2$ appears in the $\mathcal{O}(\frac{1}{T})$ term, which says that it asymptotically has no impact on the convergence. This matches the result of momentum SGD (Yu

et al., 2019a). When $T \geq \mathcal{O}(n^3)$ is sufficiently large, the third term in (2) vanishes, and the convergence rate becomes $\mathcal{O}(\frac{1}{\sqrt{nT}})$. Therefore, to reach an $\mathcal{O}(\delta)$ stationary point, one worker ($n = 1$) needs $T = \mathcal{O}(\frac{1}{\delta^2})$ iterations, while distributed training with $n$ workers requires only $T = \mathcal{O}(\frac{1}{n\delta^2})$ iterations, which is $n$ times faster than single machine training. That is, COMP-AMS has a linear speedup in terms of the number of the local workers. Such acceleration effect has also been reported for compressed SGD (Jiang & Agrawal, 2018; Zheng et al., 2019) and momentum SGD (Yu et al., 2019a) with error feedback.

## 5 EXPERIMENTS

In this section, we provide numerical results on several common datasets. Our main objective is to validate the theoretical results, and demonstrate that the proposed COMP-AMS can approach the learning performance of full-precision AMSGrad with significantly reduced communication costs.

### 5.1 DATASETS, MODELS AND METHODS

Our experiments are conducted on various image and text datasets. The MNIST (LeCun et al., 1998) contains 60000 training samples of $28 \times 18$ gray-scale hand-written digits from 10 classes, and 10000 test samples. We train MNIST with a Convolutional Neural Network (CNN), which has two convolutional layers followed by two fully connected layers with ReLu activation. Dropout is applied after the max-pooled convolutional layer with rate 0.5. The CIFAR-10 dataset (Krizhevsky & Hinton, 2009) consists of 50000 $32 \times 32$ RGB natural images from 10 classes for training and 10000 images for testing, which is trained by LeNet-5 (LeCun et al., 1998). Moreover, we also implement ResNet-18 (He et al., 2016) on this dataset. The IMDB movie review (Maas et al., 2011) is a popular binary classification dataset for sentiment analysis. Each movie review is tokenized by top-2000 most frequently appeared words and transformed into integer vectors, which is of maximal length 500. We train a Long-Short Term Memory (LSTM) network with a 32-dimensional embedding layer and 64 LSTM cells, followed by two fully connected layers before output. Cross-entropy loss is used for all the tasks. Following the classical distributed training setting, in each training iteration, data samples are uniformly randomly assigned to the workers.

We compare COMP-AMS with full-precision distributed AMSGrad, QAdam (Chen et al., 2021a) and 1BitAdam (Tang et al., 2021). For COMP-AMS, **Top-$k$** picks top 1% gradient coordinates (i.e., sparsity 0.01). QAdam and 1BitAdam both use 1-bit quantization to achieve high compression. For MNIST and CIFAR-10, the local batch size on each worker is set to be 32. For IMDB, the local batch size is 16. The hyper-parameters in COMP-AMS are set as default $\beta_1 = 0.9$, $\beta_2 = 0.999$ and $\epsilon = 10^{-8}$, which are also used for QAdam and 1BitAdam. For 1BitAdam, the epochs for warm-up training is set to be $1/20$ of the total epochs. For all methods, we tune the initial learning rate over a fine grid (see Appendix A) and report the best results averaged over three independent runs. Our experiments are performed on a GPU cluster with NVIDIA Tesla V100 cards.

### 5.2 GENERAL EVALUATION AND COMMUNICATION EFFICIENCY

The training loss and test accuracy on MNIST + CNN, CIFAR-10 + LeNet and IMDB + LSTM are reported in Figure 1. We provide more results on larger ResNet-18 model in Appendix A. On CIFAR-10, we deploy a popular decreasing learning rate schedule, where the step size $\eta$ is divided by 10 at the 40-th and 80-th epoch, respectively. We observe:

- On MNIST, all the methods can approach the training loss and test accuracy of full-precision AMSGrad. The 1BitAdam seems slightly better, but the gap is very small. On CIFAR-10, COMP-AMS with **Block-Sign** performs the best and matches AMSGrad in terms of test accuracy.

- On IMDB, COMP-AMS with **Top-$k$** has both the fastest convergence and best generalization compared with other compressed methods. This is because the IMDB text data is more sparse (with many padded zeros), where **Top-$k$** is expected to work better than sign. The 1BitAdam converges slowly. We believe one possible reason is that 1BitAdam is quite sensitive to the quality of the warm-up training. For sparse text data, the estimation of second moment $v$ is more unstable, making the strategy of freezing $v$ by warm-up less effective.

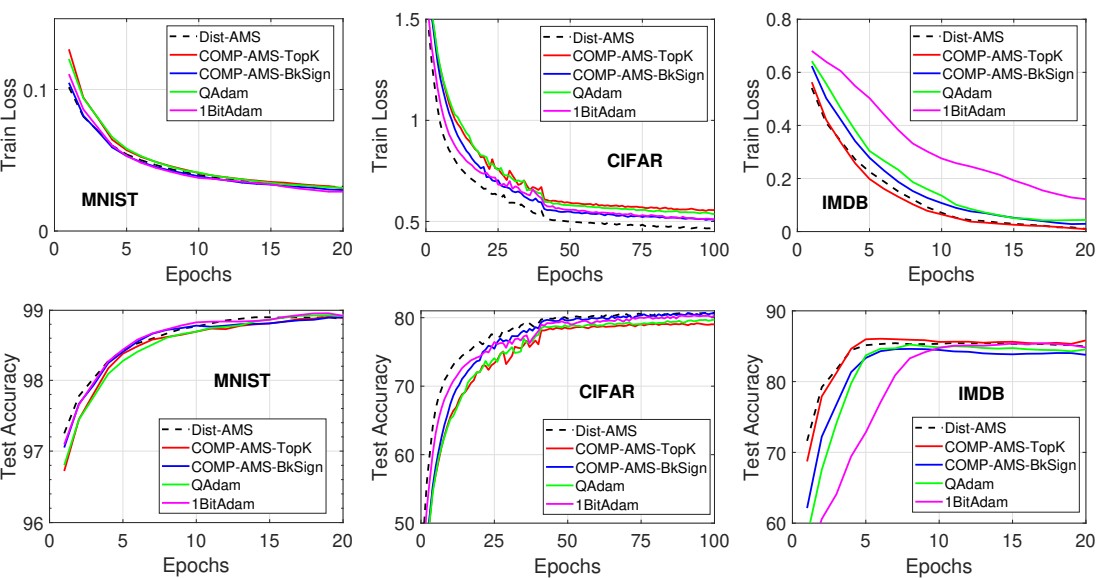

Figure 1: Training loss and test accuracy vs. epochs, on MNIST + CNN, CIFAR-10 + LeNet and IMDB + LSTM with $n = 16$ local workers.

**Communication Efficiency.** In Figure 2, we plot the training loss and test accuracy against the number of bits transmitted to the central server during the distributed training process, where we assume that the full-precision gradient is represented using 32 bits per floating number. As we can see, COMP-AMS-**Top**-0.01 achieves around 100x communication reduction, to attain similar accuracy as the full-precision distributed AMSGrad. The saving of **Block-Sign** is around 30x, but it gives slightly higher accuracy than **Top**-0.01 on MNIST and CIFAR-10. In all cases, COMP-AMS can substantially reduce the communication cost compared with full-precision distributed AMSGrad, without losing accuracy.

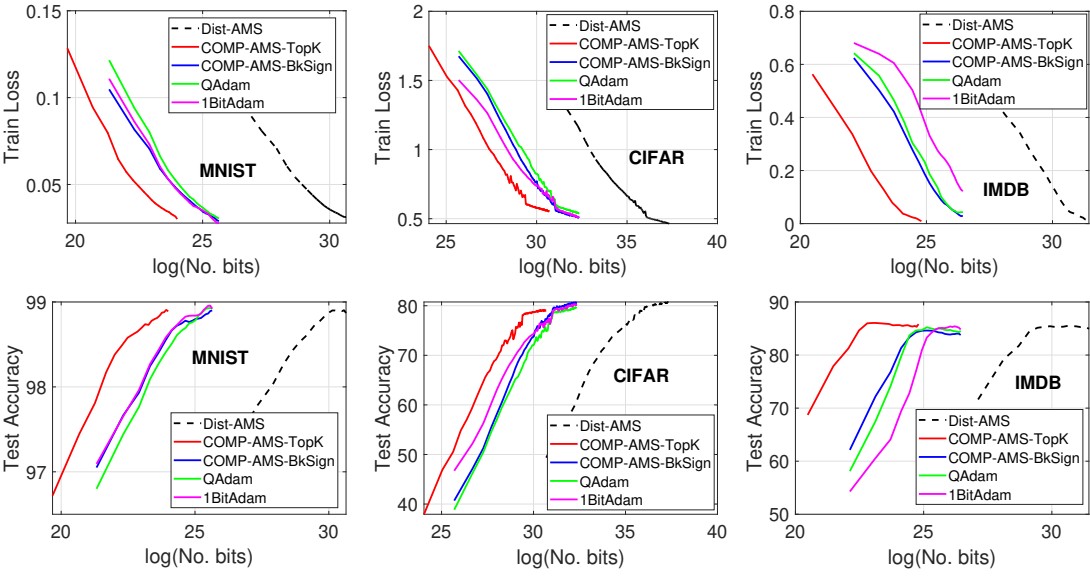

Figure 2: Train loss and Test accuracy vs. No. bits transmitted, on MNIST + CNN, CIFAR-10 + LeNet and IMDB + LSTM with $n = 16$ local workers.

## 5.3 LINEAR SPEEDUP OF COMP-AMS

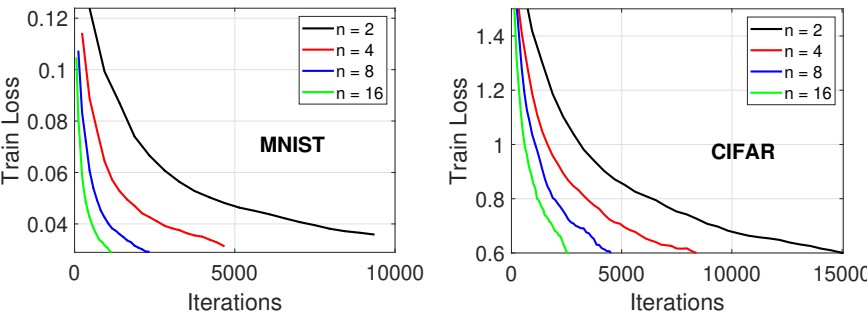

Figure 3: The linear speedup of COMP-AMS with varying $n$. **Left:** MNIST with **Block-Sign** compressor on CNN. **Right:** CIFAR-10 with **Top-$k$-0.01** compression on LeNet.

Corollary 2 reveals the linear speedup of COMP-AMS in distributed training. In Figure 3, we present the training loss on MNIST and CIFAR-10 against the number of iterations, with varying number of workers $n$. We use COMP-AMS with **Block-Sign** on MNIST, and **Top-$k$-0.01** on CIFAR. As suggested by the theory, we use $5 \times 10^{-4}\sqrt{n}$ as the learning rate. From Figure 3, we see the number of iterations to achieve a certain loss exhibits a strong linear relationship with $n$—it (approximately) decreases by half whenever we double $n$, which justifies the linear speedup of COMP-AMS.

## 5.4 DISCUSSION

We provide a brief summary of our empirical observations. The proposed COMP-AMS is able to match the learning performance of full-gradient AMSGrad in all the presented experiments. In particular, for data/model involving some sparsity structure, COMP-AMS with the **Top-$k$** compressor could be more effective. Also, our results reveal that 1BitAdam might be quite sensitive to the pre-conditioning quality, while COMP-AMS can be more easily tuned and implemented in practice.

We would like to emphasize that, the primary goal of the experiments is to show that COMP-AMS is able to match the performance of full-precision AMSGrad, but not to argue that COMP-AMS is always better than the other algorithms. Since different methods use different underlying optimization algorithms (e.g., AMSGrad, Adam, momentum SGD), comparing COMP-AMS with other distributed training methods would be largely determined by the comparison among these optimization protocols, which is typically data and task dependent. Our results say that: whenever one wants to use AMSGrad to train a deep neural network, she/he can simply employ the distributed COMP-AMS scheme to gain a linear speedup in training time with learning performance as good as the full-precision training, taking little communication cost and memory consumption.

## 6 CONCLUSION

In this paper, we study the simple, convenient, yet unexplored gradient averaging strategy for distributed adaptive optimization called COMP-AMS. **Top-$k$** and **Block-Sign** compressor are incorporated for communication efficiency, whose biases are compensated by the error feedback strategy. We develop the convergence rate of COMP-AMS, and show that same as the case of SGD, for AMSGrad, compressed gradient averaging with error feedback matches the convergence of full-gradient AMSGrad, and linear speedup can be obtained in the distributed training. Numerical experiments are conducted to justify the theoretical findings, and demonstrate that COMP-AMS provides comparable performance with other distributed adaptive methods, and achieves similar accuracy as full-precision AMSGrad with significantly reduced communication overhead. Given the simple architecture and hardware (memory) efficiency, we expect COMP-AMS shall be able to serve as a useful and convenient distributed adaptive optimization framework in practice.

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

## A    TUNING DETAILS AND MORE RESULTS ON RESNET-18

The search grids of the learning rate of each method can be found in Table 1. Empirically, Dist-AMS, COMP-AMS and 1BitAdam has similar optimal learning rate, while QAdam usually needs larger step size to reach its best performance.

Table 1: Search grids for learning rate tuning.

|  | Learning rate range |
|---|---|
| Dist-AMS | $[0.00001, 0.00003, 0.00005, 0.0001, 0.0003, 0.0005, 0.001, 0.003, 0.005, 0.01]$ |
| Comp-AMS | $[0.00001, 0.00003, 0.00005, 0.0001, 0.0003, 0.0005, 0.001, 0.003, 0.005, 0.01]$ |
| QAdam | $[0.0001, 0.0003, 0.0005, 0.001, 0.003, 0.005, 0.01, 0.03, 0.05, 0.1, 0.3, 0.5]$ |
| 1BitAdam | $[0.00001, 0.00003, 0.00005, 0.0001, 0.0003, 0.0005, 0.001, 0.003, 0.005, 0.01]$ |

We provide more experimental results on CIFAR-10 dataset, trained with ResNet-18 (He et al., 2016). For reference, we also present the result of distributed SGD. As we can see from Figure 4, again COMP-AMS can achieve similar accuracy as AMSGrad, and the **Top-**$k$ compressor gives the best accuracy, with substantial communication reduction. Note that distributed SGD converges faster than adaptive methods, but the generalization error is slightly worse. This experiment again confirms that COMP-AMS can serve as a simple and convenient distributed adaptive training framework with fast convergence, reduced communication and little performance drop.

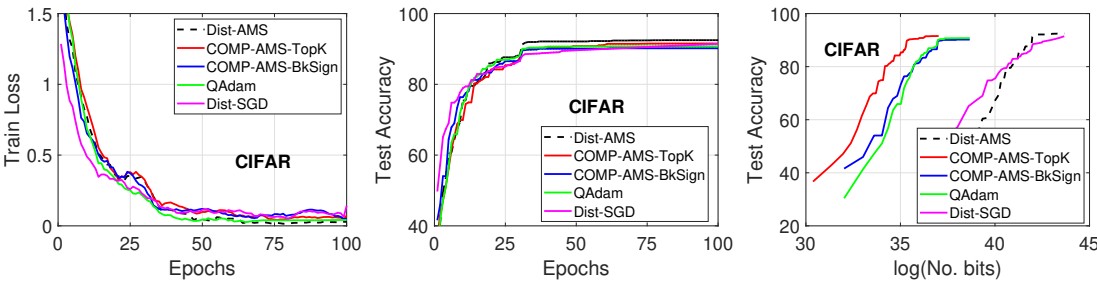

Figure 4: Training loss and test accuracy of different distributed training methods on CIFAR-10 with ResNet-18 (He et al., 2016).

# B    PROOF OF CONVERGENCE RESULTS

In this section, we provide the proof of our main result.

## B.1    PROOF OF THEOREM 1

**Theorem.** *Denote* $C_0 = \sqrt{\frac{4(1+q^2)^3}{(1-q^2)^2}G^2 + \epsilon}$, $C_1 = \frac{\beta_1}{1-\beta_1} + \frac{2q}{1-q^2}$. *Under Assumption 1 to Assumption 4, with* $\eta_t = \eta \leq \frac{\epsilon}{3C_0\sqrt{2L\max\{2L,C_2\}}}$, *for any* $T > 0$, COMP-AMS *satisfies*

$$\frac{1}{T}\sum_{t=1}^{T}\mathbb{E}[\|\nabla f(\theta_t)\|^2] \leq 2C_0\Big(\frac{\mathbb{E}[f(\theta_1) - f(\theta^*)]}{T\eta} + \frac{\eta L\sigma^2}{n\epsilon} + \frac{3\eta^2 LC_0 C_1^2 \sigma^2}{n\epsilon^2}$$

$$+ \frac{12\eta^2 q^2 LC_0 \sigma_g^2}{(1-q^2)^2\epsilon^2} + \frac{(1+C_1)G^2 d}{T\sqrt{\epsilon}} + \frac{\eta(1+2C_1)C_1 LG^2 d}{T\epsilon}\Big).$$

*Proof.* We first clarify some notations. At time $t$, let the full-precision gradient of the $i$-th worker be $g_{t,i}$, the error accumulator be $e_{t,i}$, and the compressed gradient be $\tilde{g}_{t,i} = \mathcal{C}(g_{t,i} + e_{t,i})$. Slightly different from the notations in the algorithm, we denote $\bar{g}_t = \frac{1}{n}\sum_{i=1}^{n} g_{t,i}$, $\bar{\tilde{g}}_t = \frac{1}{n}\sum_{i=1}^{n}\tilde{g}_{t,i}$ and $\bar{e}_t = \frac{1}{n}\sum_{i=1}^{n} e_{t,i}$. The second moment computed by the compressed gradients is denoted as $v_t = \beta_2 v_{t-1} + (1-\beta_2)\bar{\tilde{g}}_t^2$, and $\hat{v}_t = \max\{\hat{v}_{t-1}, v_t\}$. Also, the first order moving average sequence

$$m_t = \beta_1 m_{t-1} + (1-\beta_1)\bar{\tilde{g}}_t \quad \text{and} \quad m'_t = \beta_1 m'_{t-1} + (1-\beta_1)\bar{g}_t,$$

where $m'_t$ represents the first moment moving average sequence using the uncompressed stochastic gradients. By construction we have $m'_t = (1-\beta_1)\sum_{\tau=1}^{t}\beta_1^{t-\tau}\bar{g}_\tau$.

Our proof will use the following auxiliary sequences,

$$\mathcal{E}_{t+1} := (1-\beta_1)\sum_{\tau=1}^{t+1}\beta_1^{t+1-\tau}\bar{e}_\tau,$$

$$\theta'_{t+1} := \theta_{t+1} - \eta\frac{\mathcal{E}_{t+1}}{\sqrt{\hat{v}_t} + \epsilon}.$$

Then, we can write the evolution of $\theta'_t$ as

$$\theta'_{t+1} = \theta_{t+1} - \eta\frac{\mathcal{E}_{t+1}}{\sqrt{\hat{v}_t} + \epsilon}$$

$$= \theta_t - \eta\frac{(1-\beta_1)\sum_{\tau=1}^{t}\beta_1^{t-\tau}\bar{\tilde{g}}_\tau + (1-\beta_1)\sum_{\tau=1}^{t+1}\beta_1^{t+1-\tau}\bar{e}_\tau}{\sqrt{\hat{v}_t} + \epsilon}$$

$$= \theta_t - \eta\frac{(1-\beta_1)\sum_{\tau=1}^{t}\beta_1^{t-\tau}(\bar{\tilde{g}}_\tau + \bar{e}_{\tau+1}) + (1-\beta)\beta_1^t\bar{e}_1}{\sqrt{\hat{v}_t} + \epsilon}$$

$$= \theta_t - \eta\frac{(1-\beta_1)\sum_{\tau=1}^{t}\beta_1^{t-\tau}\bar{e}_\tau}{\sqrt{\hat{v}_t} + \epsilon} - \eta\frac{m'_t}{\sqrt{\hat{v}_t} + \epsilon}$$

$$= \theta_t - \eta\frac{\mathcal{E}_t}{\sqrt{\hat{v}_{t-1}} + \epsilon} - \eta\frac{m'_t}{\sqrt{\hat{v}_t} + \epsilon} + \eta\Big(\frac{1}{\sqrt{\hat{v}_{t-1}} + \epsilon} - \frac{1}{\sqrt{\hat{v}_t} + \epsilon}\Big)\mathcal{E}_t$$

$$\overset{(a)}{=} \theta'_t - \eta\frac{m'_t}{\sqrt{\hat{v}_t} + \epsilon} + \eta\Big(\frac{1}{\sqrt{\hat{v}_{t-1}} + \epsilon} - \frac{1}{\sqrt{\hat{v}_t} + \epsilon}\Big)\mathcal{E}_t$$

$$:= \theta'_t - \eta\frac{m'_t}{\sqrt{\hat{v}_t} + \epsilon} + \eta D_t\mathcal{E}_t,$$

where (a) uses the fact that for every $i \in [n]$, $\tilde{g}_{t,i} + e_{t+1,i} = g_{t,i} + e_{t,i}$, and $e_{t,1} = 0$ at initialization. Further define the virtual iterates:

$$x_{t+1} := \theta'_{t+1} - \eta\frac{\beta_1}{1-\beta_1}\frac{m'_t}{\sqrt{\hat{v}_t} + \epsilon},$$

which follows the recurrence:

$$
\begin{aligned}
x_{t+1} &= \theta'_{t+1} - \eta \frac{\beta_1}{1-\beta_1} \frac{m'_t}{\sqrt{\hat{v}_t} + \epsilon} \\
&= \theta'_t - \eta \frac{m'_t}{\sqrt{\hat{v}_t} + \epsilon} - \eta \frac{\beta_1}{1-\beta_1} \frac{m'_t}{\sqrt{\hat{v}_t} + \epsilon} + \eta D_t \mathcal{E}_t \\
&= \theta'_t - \eta \frac{\beta_1 m'_{t-1} + (1-\beta_1)\bar{g}_t + \frac{\beta_1^2}{1-\beta_1} m'_{t-1} + \beta_1 \bar{g}_t}{\sqrt{\hat{v}_t} + \epsilon} + \eta D_t \mathcal{E}_t \\
&= \theta'_t - \eta \frac{\beta_1}{1-\beta_1} \frac{m'_{t-1}}{\sqrt{\hat{v}_t} + \epsilon} - \eta \frac{\bar{g}_t}{\sqrt{\hat{v}_t} + \epsilon} + \eta D_t \mathcal{E}_t \\
&= x_t - \eta \frac{\bar{g}_t}{\sqrt{\hat{v}_t} + \epsilon} + \eta \frac{\beta_1}{1-\beta_1} D_t m'_{t-1} + \eta D_t \mathcal{E}_t.
\end{aligned}
$$

When summing over $t = 1, ..., T$, the difference sequence $D_t$ satisfies the bounds of Lemma 5. By the smoothness Assumption 2, we have

$$
f(x_{t+1}) \le f(x_t) + \langle \nabla f(x_t), x_{t+1} - x_t \rangle + \frac{L}{2} \|x_{t+1} - x_t\|^2.
$$

Taking expectation w.r.t. the randomness at time $t$, we obtain

$$
\mathbb{E}[f(x_{t+1})] - f(x_t)
$$

$$
\le -\eta \mathbb{E}[\langle \nabla f(x_t), \frac{\bar{g}_t}{\sqrt{\hat{v}_t} + \epsilon} \rangle] + \eta \mathbb{E}[\langle \nabla f(x_t), \frac{\beta_1}{1-\beta_1} D_t m'_{t-1} + D_t \mathcal{E}_t \rangle]
$$

$$
+ \frac{\eta^2 L}{2} \mathbb{E}[\| \frac{\bar{g}_t}{\sqrt{\hat{v}_t} + \epsilon} - \frac{\beta_1}{1-\beta_1} D_t m'_{t-1} - D_t \mathcal{E}_t \|^2]
$$

$$
= \underbrace{-\eta \mathbb{E}[\langle \nabla f(\theta_t), \frac{\bar{g}_t}{\sqrt{\hat{v}_t} + \epsilon} \rangle]}_{I} + \underbrace{\eta \mathbb{E}[\langle \nabla f(x_t), \frac{\beta_1}{1-\beta_1} D_t m'_{t-1} + D_t \mathcal{E}_t \rangle]}_{II}
$$

$$
+ \underbrace{\frac{\eta^2 L}{2} \mathbb{E}[\| \frac{\bar{g}_t}{\sqrt{\hat{v}_t} + \epsilon} - \frac{\beta_1}{1-\beta_1} D_t m'_{t-1} - D_t \mathcal{E}_t \|^2]}_{III} + \underbrace{\eta \mathbb{E}[\langle \nabla f(\theta_t) - \nabla f(x_t), \frac{\bar{g}_t}{\sqrt{\hat{v}_t} + \epsilon} \rangle]}_{IV},
$$

$$
\tag{3}
$$

In the following, we bound the terms separately.

**Bounding term I.** We have

$$
I = -\eta \mathbb{E}[\langle \nabla f(\theta_t), \frac{\bar{g}_t}{\sqrt{\hat{v}_{t-1}} + \epsilon} \rangle] - \eta \mathbb{E}[\langle \nabla f(\theta_t), (\frac{1}{\sqrt{\hat{v}_t} + \epsilon} - \frac{1}{\sqrt{\hat{v}_{t-1}} + \epsilon}) \bar{g}_t \rangle]
$$

$$
\le -\eta \mathbb{E}[\langle \nabla f(\theta_t), \frac{\nabla f(\theta_t)}{\sqrt{\hat{v}_{t-1}} + \epsilon} \rangle] + \eta G^2 \mathbb{E}[\|D_t\|].
$$

$$
\le -\frac{\eta}{\sqrt{\frac{4(1+q^2)^3}{(1-q^2)^2} G^2 + \epsilon}} \mathbb{E}[\|\nabla f(\theta_t)\|^2] + \eta G^2 \mathbb{E}[\|D_t\|_1], \tag{4}
$$

where we use Assumption 3, Lemma 4 and the fact that $l_2$ norm is no larger than $l_1$ norm.

**Bounding term II.** By the definition of $\mathcal{E}_t$, we know that $\|\mathcal{E}_t\| \le (1-\beta_1) \sum_{\tau=1}^t \beta_1^{t-\tau} \|\bar{e}_t\| \le \frac{2q}{1-q^2} G$. Then we have

$$
II \le \eta (\mathbb{E}[\langle \nabla f(\theta_t), \frac{\beta_1}{1-\beta_1} D_t m'_{t-1} + D_t \mathcal{E}_t \rangle] + \mathbb{E}[\langle \nabla f(x_t) - \nabla f(\theta_t), \frac{\beta_1}{1-\beta_1} D_t m'_{t-1} + D_t \mathcal{E}_t \rangle])
$$

$$
\le \eta \mathbb{E}[\|\nabla f(\theta_t)\| \| \frac{\beta_1}{1-\beta_1} D_t m'_{t-1} + D_t \mathcal{E}_t \|] + \eta^2 L \mathbb{E}[\| \frac{\frac{\beta_1}{1-\beta_1} m'_{t-1} + \mathcal{E}_t}{\sqrt{\hat{v}_{t-1}} + \epsilon} \| \| \frac{\beta_1}{1-\beta_1} D_t m'_{t-1} + D_t \mathcal{E}_t \|]
$$

$$
\le \eta C_1 G^2 \mathbb{E}[\|D_t\|_1] + \frac{\eta^2 C_1^2 L G^2}{\sqrt{\epsilon}} \mathbb{E}[\|D_t\|_1], \tag{5}
$$

where $C_1 := \frac{\beta_1}{1-\beta_1} + \frac{2q}{1-q^2}$. The second inequality is because of smoothness of $f(\theta)$, and the last inequality is due to Lemma 2, Assumption 3 and the property of norms.

**Bounding term III.** This term can be bounded as follows:

$$
III \leq \eta^2 L \mathbb{E}[\|\frac{\bar{g}_t}{\sqrt{\hat{v}_t}+\epsilon}\|^2] + \eta^2 L \mathbb{E}[\|\frac{\beta_1}{1-\beta_1}D_t m'_{t-1} - D_t \mathcal{E}_t\|^2]]
$$

$$
\leq \frac{\eta^2 L}{\epsilon}\mathbb{E}[\|\frac{1}{n}\sum_{i=1}^n g_{t,i} - \nabla f(\theta_t) + \nabla f(\theta_t)\|^2] + \eta^2 L \mathbb{E}[\|D_t(\frac{\beta_1}{1-\beta_1}m'_{t-1} - \mathcal{E}_t)\|^2]
$$

$$
\overset{(a)}{\leq} \frac{\eta^2 L}{\epsilon}\mathbb{E}[\|\nabla f(\theta_t)\|^2] + \frac{\eta^2 L \sigma^2}{n\epsilon} + \eta^2 C_1^2 L G^2 \mathbb{E}[\|D_t\|^2], \tag{6}
$$

where (a) follows from $\nabla f(\theta_t) = \frac{1}{n}\sum_{i=1}^n \nabla f_i(\theta_t)$ and Assumption 4 that $g_{t,i}$ is unbiased of $\nabla f_i(\theta_t)$ and has bounded variance $\sigma^2$.

**Bounding term IV.** We have

$$
IV = \eta \mathbb{E}[\langle \nabla f(\theta_t) - \nabla f(x_t), \frac{\bar{g}_t}{\sqrt{\hat{v}_{t-1}}+\epsilon}\rangle] + \eta \mathbb{E}[\langle \nabla f(\theta_t) - \nabla f(x_t), (\frac{1}{\sqrt{\hat{v}_t}+\epsilon} - \frac{1}{\sqrt{\hat{v}_{t-1}}+\epsilon})\bar{g}_t\rangle]
$$

$$
\leq \eta \mathbb{E}[\langle \nabla f(\theta_t) - \nabla f(x_t), \frac{\nabla f(\theta_t)}{\sqrt{\hat{v}_{t-1}}+\epsilon}\rangle] + \eta^2 L \mathbb{E}[\|\frac{\frac{\beta_1}{1-\beta_1}m'_{t-1} + \mathcal{E}_t}{\sqrt{\hat{v}_{t-1}}+\epsilon}\|\|D_t g_t\|]
$$

$$
\overset{(a)}{\leq} \frac{\eta\rho}{2\epsilon}\mathbb{E}[\|\nabla f(\theta_t)\|^2] + \frac{\eta}{2\rho}\mathbb{E}[\|\nabla f(\theta_t) - \nabla f(x_t)\|^2] + \frac{\eta^2 C_1 L G^2}{\sqrt{\epsilon}}\mathbb{E}[\|D_t\|]
$$

$$
\overset{(b)}{\leq} \frac{\eta\rho}{2\epsilon}\mathbb{E}[\|\nabla f(\theta_t)\|^2] + \frac{\eta^3 L}{2\rho}\mathbb{E}[\|\frac{\frac{\beta_1}{1-\beta_1}m'_{t-1} + \mathcal{E}_t}{\sqrt{\hat{v}_{t-1}}+\epsilon}\|^2] + \frac{\eta^2 C_1 L G^2}{\sqrt{\epsilon}}\mathbb{E}[\|D_t\|_1], \tag{7}
$$

where (a) is due to Young's inequality and (b) is based on Assumption 2.

Regarding the second term in (7), by Lemma 3 and Lemma 1, summing over $t = 1, ..., T$ we have

$$
\sum_{t=1}^T \frac{\eta^3 L}{2\rho}\mathbb{E}[\|\frac{\frac{\beta_1}{1-\beta_1}m'_{t-1} + \mathcal{E}_t}{\sqrt{\hat{v}_{t-1}}+\epsilon}\|^2]
$$

$$
\leq \sum_{t=1}^T \frac{\eta^3 L}{2\rho\epsilon}\mathbb{E}[\|\frac{\beta_1}{1-\beta_1}m'_{t-1} + \mathcal{E}_t\|^2]
$$

$$
\leq \sum_{t=1}^T \frac{\eta^3 L}{\rho\epsilon}\Big[\frac{\beta_1^2}{(1-\beta_1)^2}\mathbb{E}[\|m'_t\|^2] + \mathbb{E}[\|\mathcal{E}_t\|^2]\Big]
$$

$$
\leq \frac{T\eta^3 \beta_1^2 L\sigma^2}{n\rho(1-\beta_1)^2\epsilon} + \frac{\eta^3 \beta_1^2 L}{\rho(1-\beta_1)^2\epsilon}\sum_{t=1}^T \mathbb{E}[\|\nabla f(\theta_t)\|^2]
$$

$$
+ \frac{4T\eta^3 q^2 L}{\rho(1-q^2)^2\epsilon}(\sigma^2 + \sigma_g^2) + \frac{4\eta^3 q^2 L}{\rho(1-q^2)^2\epsilon}\sum_{t=1}^T \mathbb{E}[\|\nabla f(\theta_t)\|^2]
$$

$$
= \frac{T\eta^3 L C_2 \sigma^2}{n\rho\epsilon} + \frac{4T\eta^3 q^2 L\sigma_g^2}{\rho(1-q^2)^2\epsilon} + \frac{\eta^3 L C_2}{\rho\epsilon}\sum_{t=1}^T \mathbb{E}[\|\nabla f(\theta_t)\|^2], \tag{8}
$$

with $C_2 := \frac{\beta_1^2}{(1-\beta_1)^2} + \frac{4q^2}{(1-q^2)^2}$. Now integrating (4), (5), (6), (7) and (8) into (3), taking the telescoping summation over $t = 1, ..., T$, we obtain

$$
\mathbb{E}[f(x_{T+1}) - f(x_1)]
$$

$$
\leq (-\frac{\eta}{C_0} + \frac{\eta^2 L}{\epsilon} + \frac{\eta\rho}{2\epsilon} + \frac{\eta^3 L C_2}{\rho\epsilon})\sum_{t=1}^T \mathbb{E}[\|\nabla f(\theta_t)\|^2] + \frac{T\eta^2 L\sigma^2}{n\epsilon} + \frac{T\eta^3 L C_2 \sigma^2}{n\rho\epsilon} + \frac{4T\eta^3 q^2 L\sigma_g^2}{\rho(1-q^2)^2\epsilon}
$$

$$
+ (\eta(1+C_1)G^2 + \frac{\eta^2(1+C_1)C_1 L G^2}{\sqrt{\epsilon}})\sum_{t=1}^T \mathbb{E}[\|D_t\|_1] + \eta^2 C_1^2 L G^2 \sum_{t=1}^T \mathbb{E}[\|D_t\|^2].
$$

with $C_0 := \sqrt{\frac{4(1+q^2)^3}{(1-q^2)^2}G^2 + \epsilon}$. Setting $\eta \leq \frac{\epsilon}{3C_0\sqrt{2L\max\{2L,C_2\}}}$ and choosing $\rho = \frac{\epsilon}{3C_0}$, we further arrive at

$$
\mathbb{E}[f(x_{T+1}) - f(x_1)]
$$
$$
\leq -\frac{\eta}{2C_0}\sum_{t=1}^{T}\mathbb{E}[\|\nabla f(\theta_t)\|^2] + \frac{T\eta^2 L\sigma^2}{n\epsilon} + \frac{3T\eta^3 LC_0 C_2\sigma^2}{n\epsilon^2} + \frac{12T\eta^3 q^2 LC_0\sigma_g^2}{(1-q^2)^2\epsilon^2}
$$
$$
+ \frac{\eta(1+C_1)G^2 d}{\sqrt{\epsilon}} + \frac{\eta^2(1+2C_1)C_1 LG^2 d}{\epsilon}.
$$

where the inequality follows from Lemma 5. Re-arranging terms, we get that

$$
\frac{1}{T}\sum_{t=1}^{T}\mathbb{E}[\|\nabla f(\theta_t)\|^2] \leq 2C_0\Big(\frac{\mathbb{E}[f(x_1) - f(x_{T+1})]}{T\eta} + \frac{\eta L\sigma^2}{n\epsilon} + \frac{3\eta^2 LC_0 C_2\sigma^2}{n\epsilon^2}
$$
$$
+ \frac{12\eta^2 q^2 LC_0\sigma_g^2}{(1-q^2)^2\epsilon^2} + \frac{(1+C_1)G^2 d}{T\sqrt{\epsilon}} + \frac{\eta(1+2C_1)C_1 LG^2 d}{T\epsilon}\Big)
$$
$$
\leq 2C_0\Big(\frac{\mathbb{E}[f(\theta_1) - f(\theta^*)]}{T\eta} + \frac{\eta L\sigma^2}{n\epsilon} + \frac{3\eta^2 LC_0 C_1^2\sigma^2}{n\epsilon^2}
$$
$$
+ \frac{12\eta^2 q^2 LC_0\sigma_g^2}{(1-q^2)^2\epsilon^2} + \frac{(1+C_1)G^2 d}{T\sqrt{\epsilon}} + \frac{\eta(1+2C_1)C_1 LG^2 d}{T\epsilon}\Big),
$$

where $C_0 = \sqrt{\frac{4(1+q^2)^3}{(1-q^2)^2}G^2 + \epsilon}$, $C_1 = \frac{\beta_1}{1-\beta_1} + \frac{2q}{1-q^2}$. The last inequality is because $x_1 = \theta_1$, $\theta^* := \arg\min_\theta f(\theta)$ and the fact that $C_2 \leq C_1^2$. This completes the proof. $\qquad\square$

## B.2 INTERMEDIATE LEMMATA

The lemmas used in the proof of Theorem 1 are given as below.

**Lemma 1.** *Under Assumption 1 to Assumption 4 we have:*

$$
\|m'_t\| \leq G, \quad \forall t,
$$
$$
\sum_{t=1}^{T}\mathbb{E}\|m'_t\|^2 \leq \frac{T\sigma^2}{n} + \sum_{t=1}^{T}\mathbb{E}[\|\nabla f(\theta_t)\|^2].
$$

*Proof.* For the first part, it is easy to see that by Assumption 3,

$$
\|m'_t\| = (1-\beta_1)\|\sum_{\tau=1}^{t}\beta_1^{t-\tau}\bar{g}_t\| \leq G.
$$

For the second claim, the expected squared norm of average stochastic gradient can be bounded by

$$
\mathbb{E}[\|\bar{g}_t^2\|] = \mathbb{E}[\|\frac{1}{n}\sum_{i=1}^{n}g_{t,i} - \nabla f(\theta_t) + \nabla f(\theta_t)\|^2]
$$
$$
= \mathbb{E}[\|\frac{1}{n}\sum_{i=1}^{n}(g_{t,i} - \nabla f_i(\theta_t))\|^2] + \mathbb{E}[\|\nabla f(\theta_t)\|^2]
$$
$$
\leq \frac{\sigma^2}{n} + \mathbb{E}[\|\nabla f(\theta_t)\|^2],
$$

where we use Assumption 4 that $g_{t,i}$ is unbiased with bounded variance. Let $\bar{g}_{t,j}$ denote the $j$-th coordinate of $\bar{g}_t$. By the updating rule of COMP-AMS, we have

$$
\begin{aligned}
\mathbb{E}[\|m_t'\|^2] &= \mathbb{E}[\|(1-\beta_1)\sum_{\tau=1}^{t}\beta_1^{t-\tau}\bar{g}_\tau\|^2] \\
&\leq (1-\beta_1)^2 \sum_{j=1}^{d}\mathbb{E}[(\sum_{\tau=1}^{t}\beta_1^{t-\tau}\bar{g}_{\tau,j})^2] \\
&\overset{(a)}{\leq} (1-\beta_1)^2 \sum_{j=1}^{d}\mathbb{E}[(\sum_{\tau=1}^{t}\beta_1^{t-\tau})(\sum_{\tau=1}^{t}\beta_1^{t-\tau}\bar{g}_{\tau,j}^2)] \\
&\leq (1-\beta_1)\sum_{\tau=1}^{t}\beta_1^{t-\tau}\mathbb{E}[\|\bar{g}_\tau\|^2] \\
&\leq \frac{\sigma^2}{n} + (1-\beta_1)\sum_{\tau=1}^{t}\beta_1^{t-\tau}\mathbb{E}[\|\nabla f(\theta_t)\|^2],
\end{aligned}
$$

where (a) is due to Cauchy-Schwartz inequality. Summing over $t = 1,...,T$, we obtain

$$
\sum_{t=1}^{T}\mathbb{E}\|m_t'\|^2 \leq \frac{T\sigma^2}{n} + \sum_{t=1}^{T}\mathbb{E}[\|\nabla f(\theta_t)\|^2].
$$

This completes the proof. $\square$

**Lemma 2.** *Under Assumption 4, we have for $\forall t$ and each local worker $\forall i \in [n]$,*

$$
\|e_{t,i}\|^2 \leq \frac{4q^2}{(1-q^2)^2}G^2,
$$

$$
\mathbb{E}[\|e_{t+1,i}\|^2] \leq \frac{4q^2}{(1-q^2)^2}\sigma^2 + \frac{2q^2}{1-q^2}\sum_{\tau=1}^{t}(\frac{1+q^2}{2})^{t-\tau}\mathbb{E}[\|\nabla f_i(\theta_\tau)\|^2].
$$

*Proof.* We start by using Assumption 1 and Young's inequality to get

$$
\begin{aligned}
\|e_{t+1,i}\|^2 &= \|g_{t,i} + e_{t,i} - \mathcal{C}(g_{t,i} + e_{t,i})\|^2 \\
&\leq q^2\|g_{t,i} + e_{t,i}\|^2 \\
&\leq q^2(1+\rho)\|e_{t,i}\|^2 + q^2(1+\frac{1}{\rho})\|g_{t,i}\|^2 \\
&\leq \frac{1+q^2}{2}\|e_{t,i}\|^2 + \frac{2q^2}{1-q^2}\|g_{t,i}\|^2,
\end{aligned} \tag{9}
$$

where (9) is derived by choosing $\rho = \frac{1-q^2}{2q^2}$ and the fact that $q < 1$. Now by recursion and the initialization $e_{1,i} = 0$, we have

$$
\begin{aligned}
\mathbb{E}[\|e_{t+1,i}\|^2] &\leq \frac{2q^2}{1-q^2}\sum_{\tau=1}^{t}(\frac{1+q^2}{2})^{t-\tau}\mathbb{E}[\|g_{\tau,i}\|^2] \\
&\leq \frac{4q^2}{(1-q^2)^2}\sigma^2 + \frac{2q^2}{1-q^2}\sum_{\tau=1}^{t}(\frac{1+q^2}{2})^{t-\tau}\mathbb{E}[\|\nabla f_i(\theta_\tau)\|^2],
\end{aligned}
$$

which proves the second argument. Meanwhile, the absolute bound $\|e_{t,i}\|^2 \leq \frac{4q^2}{(1-q^2)^2}G^2$ follows directly from (9). $\square$

**Lemma 3.** *For the moving average error sequence $\mathcal{E}_t$, it holds that*

$$
\sum_{t=1}^{T}\mathbb{E}[\|\mathcal{E}_t\|^2] \leq \frac{4Tq^2}{(1-q^2)^2}(\sigma^2 + \sigma_g^2) + \frac{4q^2}{(1-q^2)^2}\sum_{t=1}^{T}\mathbb{E}[\|\nabla f(\theta_t)\|^2].
$$

*Proof.* Denote $K_{t,i} := \sum_{\tau=1}^{t}(\frac{1+q^2}{2})^{t-\tau}\mathbb{E}[\|\nabla f_i(\theta_\tau)\|^2]$. Using the same technique as in the proof of Lemma 1, denoting $\bar{e}_{t,j}$ as the $j$-th coordinate of $\bar{e}_t$, it follows that

$$\mathbb{E}[\|\mathcal{E}_t\|^2] = \mathbb{E}[\|(1-\beta_1)\sum_{\tau=1}^{t}\beta_1^{t-\tau}\bar{e}_\tau\|^2]$$

$$\leq (1-\beta_1)^2\sum_{j=1}^{d}\mathbb{E}[(\sum_{\tau=1}^{t}\beta_1^{t-\tau}\bar{e}_{\tau,j})^2]$$

$$\overset{(a)}{\leq} (1-\beta_1)^2\sum_{j=1}^{d}\mathbb{E}[(\sum_{\tau=1}^{t}\beta_1^{t-\tau})(\sum_{\tau=1}^{t}\beta_1^{t-\tau}\bar{e}_{\tau,j}^2)]$$

$$\leq (1-\beta_1)\sum_{\tau=1}^{t}\beta_1^{t-\tau}\mathbb{E}[\|\bar{e}_\tau\|^2]$$

$$\leq (1-\beta_1)\sum_{\tau=1}^{t}\beta_1^{t-\tau}\mathbb{E}[\frac{1}{n}\sum_{i=1}^{n}\|e_{\tau,i}\|^2]$$

$$\overset{(b)}{\leq} \frac{4q^2}{(1-q^2)^2}\sigma^2 + \frac{2q^2(1-\beta_1)}{(1-q^2)}\sum_{\tau=1}^{t}\beta_1^{t-\tau}(\frac{1}{n}\sum_{i=1}^{n}K_{\tau,i}),$$

where (a) is due to Cauchy-Schwartz and (b) is a result of Lemma 2. Summing over $t = 1, ..., T$ and using the technique of geometric series summation leads to

$$\sum_{t=1}^{T}\mathbb{E}[\|\mathcal{E}_t\|^2] \leq \frac{4Tq^2}{(1-q^2)^2}\sigma^2 + \frac{2q^2(1-\beta_1)}{(1-q^2)}\sum_{t=1}^{T}\sum_{\tau=1}^{t}\beta_1^{t-\tau}(\frac{1}{n}\sum_{i=1}^{n}K_{\tau,i})$$

$$\leq \frac{4Tq^2}{(1-q^2)^2}\sigma^2 + \frac{2q^2}{(1-q^2)}\sum_{t=1}^{T}\sum_{\tau=1}^{t}(\frac{1+q^2}{2})^{t-\tau}\mathbb{E}[\frac{1}{n}\sum_{i=1}^{n}\|\nabla f_i(\theta_\tau)\|^2]$$

$$\leq \frac{4Tq^2}{(1-q^2)^2}\sigma^2 + \frac{4q^2}{(1-q^2)^2}\sum_{t=1}^{T}\mathbb{E}[\frac{1}{n}\sum_{i=1}^{n}\|\nabla f_i(\theta_t)\|^2]$$

$$\overset{(a)}{\leq} \frac{4Tq^2}{(1-q^2)^2}\sigma^2 + \frac{4q^2}{(1-q^2)^2}\sum_{t=1}^{T}\mathbb{E}[\|\frac{1}{n}\sum_{i=1}^{n}\nabla f_i(\theta_t)\|^2 + \frac{1}{n}\sum_{i=1}^{n}\|\nabla f_i(\theta_t) - \nabla f(\theta_t)\|^2]$$

$$\leq \frac{4Tq^2}{(1-q^2)^2}(\sigma^2 + \sigma_g^2) + \frac{4q^2}{(1-q^2)^2}\sum_{t=1}^{T}\mathbb{E}[\|\nabla f(\theta_t)\|^2],$$

where (a) is derived by the variance decomposition and the last inequality holds due to Assumption 4. The desired result is obtained. $\qquad\square$

**Lemma 4.** *It holds that* $\forall t \in [T], \forall i \in [d], \hat{v}_{t,i} \leq \frac{4(1+q^2)^3}{(1-q^2)^2}G^2$.

*Proof.* For any $t$, by Lemma 2 and Assumption 3 we have

$$\|\tilde{g}_t\|^2 = \|\mathcal{C}(g_t + e_t)\|^2$$

$$\leq \|\mathcal{C}(g_t + e_t) - (g_t + e_t) + (g_t + e_t)\|^2$$

$$\leq 2(q^2 + 1)\|g_t + e_t\|^2$$

$$\leq 4(q^2 + 1)(G^2 + \frac{4q^2}{(1-q^2)^2}G^2)$$

$$= \frac{4(1+q^2)^3}{(1-q^2)^2}G^2.$$

It's then easy to show by the updating rule of $\hat{v}_t$, there exists a $j \in [t]$ such that $\hat{v}_t = v_j$. Then

$$\hat{v}_{t,i} = (1-\beta_2)\sum_{\tau=1}^{j}\beta_2^{j-\tau}\tilde{g}_{\tau,i}^2 \leq \frac{4(1+q^2)^3}{(1-q^2)^2}G^2,$$

which concludes the claim. □

**Lemma 5.** *Let $D_t := \frac{1}{\sqrt{\hat{v}_{t-1}+\epsilon}} - \frac{1}{\sqrt{\hat{v}_t+\epsilon}}$ be defined as above. Then,*

$$\sum_{t=1}^{T} \|D_t\|_1 \le \frac{d}{\sqrt{\epsilon}}, \quad \sum_{t=1}^{T} \|D_t\|^2 \le \frac{d}{\epsilon}.$$

*Proof.* By the updating rule of COMP-AMS, $\hat{v}_{t-1} \le \hat{v}_t$ for $\forall t$. Therefore, by the initialization $\hat{v}_0 = 0$, we have

$$\sum_{t=1}^{T} \|D_t\|_1 = \sum_{t=1}^{T} \sum_{i=1}^{d} \left( \frac{1}{\sqrt{\hat{v}_{t-1,i}+\epsilon}} - \frac{1}{\sqrt{\hat{v}_{t,i}+\epsilon}} \right)$$

$$= \sum_{i=1}^{d} \left( \frac{1}{\sqrt{\hat{v}_{0,i}+\epsilon}} - \frac{1}{\sqrt{\hat{v}_{T,i}+\epsilon}} \right)$$

$$\le \frac{d}{\sqrt{\epsilon}}.$$

For the sum of squared $l_2$ norm, note the fact that for $a \ge b > 0$, it holds that

$$(a-b)^2 \le (a-b)(a+b) = a^2 - b^2.$$

Thus,

$$\sum_{t=1}^{T} \|D_t\|^2 = \sum_{t=1}^{T} \sum_{i=1}^{d} \left( \frac{1}{\sqrt{\hat{v}_{t-1,i}+\epsilon}} - \frac{1}{\sqrt{\hat{v}_{t,i}+\epsilon}} \right)^2$$

$$\le \sum_{t=1}^{T} \sum_{i=1}^{d} \left( \frac{1}{\hat{v}_{t-1,i}+\epsilon} - \frac{1}{\hat{v}_{t,i}+\epsilon} \right)$$

$$\le \frac{d}{\epsilon},$$

which gives the desired result. □

