# OpenReview forum: "On Distributed Adaptive Optimization with Gradient Compression"
_ICLR.cc/2022/Conference — ICLR 2022 Poster_

### Official Review · Reviewer_7SPd · 2021-11-02

**Correctness:** 4
**Technical Novelty And Significance:** 3
**Empirical Novelty And Significance:** 3
**Recommendation:** 5
**Confidence:** 3

**Main Review:**

Strength:
The authors extend the adaptive optimization framework to distributed approach with a compressed gradient averaging. The convergence analysis implies a linear speedup in terms of the number of workers, and shows that in the single-machine case, it can achieve the same convergence rate as the standard full-gradient SGD. The paper is well-written and easy to follow.

Weakness:
I have some concerns as follows.
1. The authors do not assume the finite-sum of $f_i(\theta)$ to be convex, then how to avoid converging to a local optimal?
2. If data heterogeneity exists for local workers, is it possible that the local gradient may drag others back when taking the gradient averaging in the central server?
3. In Algorithm 2, how to determine parameters $\beta_1, \beta_2$ and $k$? Is it possible that $k$ differs for different local workers?
4. In the algorithm, when applying EF technique to reduce the bias, each local worker still needs extra storage for the error term $e_t$, which is the same dimension as the local gradient. Why such an approach can be efficient in terms of the memory space when training large-scale learners?
5. In the experiments, how many workers are considered? Since data samples are randomly assigned to the workers, it can be regarded as the non-data heterogeneous case, it could be better if the authors can add the results for cases with data heterogeneity, e.g., workers with different sample sizes/ different class distributions.
6. There seem to be some typos. For example, in algorithm 1, line 8, I suppose the numerator should be $m_t$ instead of $\theta_t$; in theorem 1, corollary 1 and 2, what is $C_2$ in the denominator of the constrain for $\eta$?



**Summary Of The Paper:**

The authors propose the COMP-AMS algorithm for a distributed optimization framework. The algorithm is based on gradient averaging and adaptive algorithms. The application of gradient compression helps to reduce the communication complexity, and the tool of error feedback is used for the bias correction. The authors study the convergence rate of the proposed algorithm. The theoretical results are justified by the numerical experiments.

**Summary Of The Review:**

I believe the paper has its value in extending the adaptive optimization framework to distributed approach with compressed gradient averaging. I mainly have concerns about the assumption on convexity and the impact on the performance brought by data heterogeneity.

---

> ### Author Response · Authors · 2021-11-19
> **Response to Reviewer 7SPd**
>
> Dear Reviewer,
>
> Thanks for your valuable feedback.
>
> 1. In this paper, we choose to focus on the non-convex objective mainly for two reasons:
>
> (i) Practically, compression methods are most important in scenarios when training large deep neural networks which are highly non-convex.
>
> (ii) Considering non-convex loss is standard in prior works, especially in recent years on the optimization of deep networks (please see our references in Section 1 and 2). To some extent, this is also consistent with (i) since modern large-scale training objectives are mostly non-convex.
>
> In the context of non-convex optimization, the LHS in Theorem 1 (the squared magnitude of the gradient) is a standard measure of convergence. Theorem 1 characterizes how fast the LHS decreases, hence how fast the algorithm reaches a stationary point. As you kindly pointed out, this solution might be a sub-optimal point (actually it is usually true for deep neural nets, where this point might be a saddle point or a local minima). However, it has been argued based on empirical observations that the local optima in deep networks are usually close to being globally optimal (i.e., many local optima are ''good'' ones). There are also some related theoretical justifications considering the landscape of the loss function of deep learning, e.g., [1,2] and references therein. In general, we would say that this question is still not fully answered for practical deep model, and is an interesting independent topic worth further investiagtion.
>
> [1] - The Loss Surface of Deep and Wide Neural Networks, Nguyen and Hein, ICML 2017.
>
> [2] - The Global Landscape of Neural Networks: An Overview, Sun et al., 2020.
>
> 2. Yes, different local data distributions may slow down the convergence of the global model and hurt the generalisation. Yet, data heterogeneity is often considered in practice under the federated learning setting, allowing local model updates on a large number of clients (eg. personal devices). The problem setting of our work mainly follows the classical distributed optimization problem (same as most references in Section 2.1), where data stored at a central server is uniformly assigned to local workers (GPUs) at each iteration. Hence, as we stated in the first paragraph on page 6, we consider the effect of data heterogeneity (the global variance) in our theoretical analysis mainly to shed some light on the federated setting for broader interest and future investigation.
>
> 3. In most numerical results of adaptive methods in the literature we are aware of, the default $\beta_1=0.9$ and $\beta_2=0.999$ usually give very good empirical performance. Since the values of these parameters would not affect our theoretical analysis, in this work we assume them to be fixed. The sparsity level $k$ of the Top-$k$ compressor can be chosen based on the application scenario and human expertise in practice. In principle, $k$ could be different across workers. We can modify Assumption 1 to define $q_i,i=1,...,n$ for each node, which will show up in the convergence rate. However, this will not change the order of the rates. Hence, for convenience we assume same $k$ for all workers.
>
> 4. The goal of our paper (as well as the rich literature on compressed distributed training) is to significantly reduce the communication cost in practice, since transmitting the gradient of large deep nets between the central server and local workers could be slow. It is known that the error accumulator is crucial for the good performance of 1-bit quantization and top-$k$. Thus, this EF tensor is a necessary part of such algorithms (e.g., compressed SGD, our COMP-AMS, 1BitAdam and QAdam). In some sense, we trade some local memory for lower communication. Nevertheless, our COMP-AMS does not need any extra tensor other than the error accumulator, while 1BitAdam and QAdam both need more tensors locally. Since these tensors have the same size as the model, they would cost more memory on the local GPUs. From this perspective, our method is simpler and more hardware-efficient.
>
> 5. In Figure 1 and Figure 2 (see the captions), we use $n=16$ workers. In Figure 4 (CIFAR-10 trained by ResNet-18), $n=8$. As we mentioned in our answer to question 2, data heterogeneity is usually experimented in federated learning problems as it portrays more accurately the heterogeneity across a cohort of devices, eg. mobile phones. Following prior related works, our experiments are under the standard distributed training setting with i.i.d. data.
>
> 6. In Algorithm 1 line 8, the numerator should be $m_t$. In the theory, $C_2$ should be $C_1$. We have corrected these typos in the updated version. Thanks for carefully pointing them out.
>
> Again, thanks for your valuable comments. We hope our rebuttal can well address your questions.

---

### Official Review · Reviewer_c4KT · 2021-11-02

**Correctness:** 4
**Technical Novelty And Significance:** 3
**Empirical Novelty And Significance:** 3
**Recommendation:** 8
**Confidence:** 2

**Main Review:**

Major strengths are listed as follows:
(1) The paper is easy to follow and provides sufficient background on compressed gradients and adaptive optimization. Comparisons with two related works are important and clear.
(2) The proposed error-feedback strategy is interesting in the distributed optimization scenario.
(3) Empirical results on some benchmark data sets are supportive and convincing.

In addition, there are several minor issues that the authors could address to improve the clarity and quality of the paper:
1. In Sec 2.2, it seems there is no problem description for Algorithm 1, which may cause confusion for first-time readers. The paragraph in the beginning of Sec 3 could be moved here or even before Sec 2.2. Also, the parameter space could be explained or detailed with an example.
2. In Sec 2.2., "different learning rate"->"different learning rates", "element-wisely"->"elementwise", "previous gradient magnitude"->"previous gradient magnitudes".
3. In Definition 2, $sign(x_{\mathcal{B}_1})$ is not defined.
4. In Theorem 1, "Under Algorithm 1 to Algorithm 4" -> "Under Algorithms 1-4".

**Summary Of The Paper:**

The paper proposes a new distributed optimization framework-COMP-AMS-based on gradient averaging and compression with convergence guarantee. In particular, theoretical discussions have shown that the proposed algorithm shares the same convergence rate as AMSGrad with linear speedup effect. Numerical experiments on several real-world data sets have demonstrated the proposed method can significantly reduce the communication costs while reaching comparable performance with full-precision AMSGrad.

**Summary Of The Review:**

Overall, the proposed work is important and interesting, and the paper deserves publication. The compressed gradient averaging technique with error feedback and its related convergence discussions could shed lights on other related works.

---

> ### Author Response · Authors · 2021-11-19
> **Response to Reviewer c4KT**
>
> Dear Reviewer,
>
> We sincerely thank your nice summary of our contributions and the support of our work!
>
> 1. Thanks for the great suggestion. In the updated version, we have moved the problem setting to the beginning of the paper. We have also clarified the parameter space as $R^d$, following many prior works.
>
> 2. Thanks for pointing our the typos. We have corrected them, as well as the definition of $x_{B_1}$ in the revision.

---

### Official Review · Reviewer_ou6Y · 2021-11-09

**Correctness:** 4
**Technical Novelty And Significance:** 3
**Empirical Novelty And Significance:** 3
**Recommendation:** 8
**Confidence:** 3

**Main Review:**

Strengths
- The topic considering communication efficiency for adaptive distributed optimization is of timeliness and importance.
- Theoretical results look promising and the numeral results efficiently supports the validity of the theoretical results in practice.
- The paper is well-organized and contains the introduction of related studies with a fairly enough coverage.

Weaknesses:
Minor comments
- Is there any systematic ways to choose the parameters $\beta_1, \beta_2, \epsilon$ in the proposed algorithm (Algorithm 2)?
- Page 4, "In Section 4 & 5" --> "In Sections 4 and 5"
- How do we define the "data heterogeneity" in page 2? Does this mean different $\chi_i$ on each local node? It would be better if the authors could clearly state this.
- Page 6, "under Assumption 1 to Assumption 4" --> "under Assumptions 1 to 4"


**Summary Of The Paper:**

This paper suggests a gradient averaging strategy for distributed adaptive optimization. Gradient compression is used for reducing the communication costs in transmission of gradients, and the tool of error feedback is used for correcting bias injected by the compression step. Convergence analysis results demonstrates that the proposed strategy has a linear speedup as the number of workers increases while the same convergence rate as standard AMSGrad. The experimental results on real-world datasets successfully validate the theoretical results.

**Summary Of The Review:**

Overall, I believe that this paper could be a meaningful add to the theories behind the distributed optimization. Numerical results also look solid enough to support the theoretical results.

---

> ### Author Response · Authors · 2021-11-19
> **Response to Reviewer ou6Y**
>
> Dear Reviewer,
>
> We sincerely appreciate your recognition of our contributions and the support of our work! We have corrected the typo in the revised pdf file.
>
> 1. Yes, there are several approaches to choose hyper-parameters in deep network training, for example, the Gaussian process based Bayesian optimization and bandits [1,2,3]. Yet, in many of the papers using Adam or AMSGrad that we are aware of, people are simply using the default values ($\beta_1=0.9,\beta_2=0.999,\epsilon=10^{-8}$) and getting good empirical performance. Since the specific choice of these parameters would not affect our theoretical analysis, in this work we do not focus on the structured tuning of the hyper-parameters and assume them to be fixed.
>
> 2. Yes, data heterogeneity means that $\chi_i$ is different across workers, i.e., the distribution of the data that each local worker (device) received is different. We have clarified this point in the revision. Thank you.
>
> [1] - Algorithms for Hyper-Parameter Optimization,  Bergstra et al., NIPS 2011.
>
> [2] - Practical Bayesian Optimization of Machine Learning Algorithms, Snoek et al., NIPS 2012.
>
> [3] - Hyperband: A Novel Bandit-Based Approach to Hyperparameter Optimization, Li et al., JMLR 2018.

---

### Decision · Program_Chairs · 2022-01-20

**Decision:**

Accept (Poster)

**Comment:**

The paper considers the setting of distributed optimization and proposes an adaptive gradient averaging and compression scheme to reduce the communication cost. The proposed scheme is shown to achieve the same convergence rate as full-gradient AMSGrad algorithm, but due to the reduced cost, it exhibits linear speedup as the number of workers grows.

The reviews appreciated the clear presentation of the results, technical soundness, and convincing numerical experiments. The paper is a solid contribution to distributed optimization. Thus, I recommend acceptance.